# The association between maternal infection and intellectual disability in children: A systematic review and meta-analysis

Mahroo Rezaeinejad[1], Seyed Mohammad Riahi[2], Kimia Behzad Moghadam[3], Mehrdad Jafari Tadi[4], Zahra Geraili[5], Hamid Parsa[6], Elika Marhoommirzabak[6], Malihe Nourollahpour Shiadeh [7]*, Ali Alizadeh Khatir[8]*

1 Department of Obstetrics and Gynecology, Imam Khomeini Hospital Complex, Tehran University of Medical Sciences, Tehran, Iran, 2 Department of Epidemiology and Biostatistics, Cardiovascular Diseases Research Center, School of Medicine, Birjand University of Medical Sciences, Birjand, Iran, 3 Independent Researcher, Former University of California, San Francisco (UCSF), San Francisco, California, United States of America, 4 Department of Cell and Molecular Medicine, Rush University Medical Center, Chicago, Illinois, United States of America, 5 Social Determinants of Health Research Center, Health Research Institute, Babol University of Medical Sciences, Babol, Iran, 6 Department of Neurology, University of Visayas, Gullas College of Medicine, Cebu City, Philippines, 7 Sexual and Reproductive Health Research Center, Mazandaran University of Medical Sciences, Sari, Iran, 8 Health Research Institute, Mobility Impairment Research Center, Babol University of Medical Sciences, Babol, Iran

* malihe.nurollahpur@gmail.com (MNS); alizade.ali83@yahoo.com (AAK)

## Abstract

### Background

There is arguing evidence regarding the association between maternal infections during pregnancy and the risk of intellectual disability (ID) in children. This systematic review and meta-analysis are essential to determine and address inconsistent findings between maternal infections during pregnancy and the risk of ID in children.

### Methods

The MOOSE and PRISMA guidelines were followed to perform and report on this study. The Medline/PubMed, Web of Science, Embase, and Scopus databases were searched from inception up to March 15, 2023, to identify potentially eligible studies. Inclusion and exclusion criteria were applied, as well as the Newcastle-Ottawa Scale was used to assess the methodological quality of studies included. The included studies were divided into two types based on the participants: (1) ID-based studies, which involved children with ID as cases and healthy children as controls and evaluated maternal infection in these participants; (2) infection-based studies, which assessed the prevalence or incidence of ID in the follow-up of children with or without exposure to maternal infection. We used Random-effects models (REM) to estimate the overall pooled odds ratio (OR) and 95% confidence intervals (CIs). The between-studies heterogeneity was assessed with the $\chi^2$-based Q-test and $I^2$ statistic. Subgroup and sensitivity analyses were applied to explore the source of heterogeneity and results consistency.

**Data Availability Statement:** All relevant data are within the manuscript and its Supporting information files.

**Funding:** The author(s) received no specific funding for this work.

**Competing interests:** The authors have declared that no competing interests exist.

## Results

A total of eight studies including 1,375,662 participants (60,479 cases and 1,315,183 controls) met the eligibility criteria. The REM found that maternal infection significantly increased the risk of ID in children (OR, 1.33; 95% CI, 1.21–1.46; $I^2$ = 64.6). Subgroup analysis showed a significant association for both infection-based (OR, 1.27; 95%CI, 1.15–1.40; $I^2$ = 51.2) and ID-based (OR, 1.44; 95%CI, 1.19–1.74; $I^2$ = 77.1) studies. Furthermore, subgroup analysis based on diagnostic criteria revealed a significant association when maternal infection or ID were diagnosed using ICD codes (OR, 1.33; 95% CI, 1.20–1.48; $I^2$ = 75.8).

## Conclusion

Our study suggests that maternal infection during pregnancy could be associated with an increased risk of ID in children. This finding is consistent across different types of studies and diagnostic criteria. However, due to the heterogeneity and limitations of the included studies, we recommend further longitudinal studies to confirm the causal relationship and the underlying mechanisms.

## Introduction

Intellectual disability (ID) is a lifelong abnormality characterized by an IQ below 70 and deficits in both intellectual functioning and adaptive abilities, mostly identified in childhood or adolescence [1]. ID not only affects the people who suffer from it, but also considered as a family and social health problem [2]. The global prevalence of ID is between 1% and 3% in the general population with some regional variations [1]. According to the Global Burden of Disease Study, out of the 2.6 billion children and adolescents in 2017, ~83.2 million were affected by ID, representing a global prevalence of 3.2% (95% UI: 2.5%–3.9%) [3]. According to global estimates, ID accounted for approximately 10.7 million years of living with disability (YLD) in people aged below 20 years [3].

The main etiology of ID is still not well understood. While genetics or chromosomal factors are established to be the cause of ID in approximately half of cases, there is also some strong evidence for the role of environmental factors during pregnancy and embryonic neurodevelopment as causes of ID, including maternal infections [4,5]. Maternal infections are well-established risk factors for the neurological and behavioral abnormalities in children, mainly due to the teratogenic effect of neurotropic infectious agents such as rubella, cytomegalovirus or *Toxoplasma gondii* on the fetal brain [6,7]. The neurodevelopmental abnormalities associated with maternal infections could be due to brain damage directly induced by the infectious agents, or host immune responses to infection and subsequent inflammation, or indirectly by adverse birth outcomes induced by infections such as preterm birth, low birthweight, or neonatal brain injury [8–11].

There have been substantial epidemiological studies of the associations between maternal infection and neurodevelopmental abnormalities (e.g., autism, schizophrenia and cerebral palsy) [12–14], but very few studies have been done on the causal relationship between maternal infection and ID, with a degree of controversy in the results [15–17]. Moreover, no study has yet been carried out to systematically review existing data on this topic. A better knowledge of this issue may have important public health and clinical implications. Therefore, we designed and performed the present systematic review and meta-analysis and scrutinized all

publicly available observational studies to calculate the risk of ID in children who are born after being exposed to maternal infection.

## Methods

We used the Meta-Analysis of Observational Studies in Epidemiology (MOOSE) and Preferred Reporting Items for Systematic Reviews and Meta-Analyses Protocols (PRISMA) guidelines to perform and report of this study, respectively [18,19].

### Search strategy and study selection

To identify the relevant studies, two independent investigators (M.R. and H.P.) systematically explored published literature in the online scientific databases including, Medline through PubMed, Web of science collection, Embase, and SCOPUS from the date of inception until 15 March 2023 (S1 Fig). The search strategy was designed and we adapted it to each database by a medical library expert (Z.G). The following keywords were used in the databases search: "maternal infection" OR "gestational infection" OR "infection" OR "infectious" AND "mental retardation" OR "intellectual disability" OR "intellectual development" OR "development disorders" OR "mental deficiency". Furthermore, two investigators (M.J. and M.J.) searched the first 20 pages of the Google Scholar engine using the above keywords and the bibliographies of retrieved articles to find additional relevant studies and gray literature. There was no time, geographical or language limitation, although we limited literature search to "human-subjects" studies. Retrieved studies from different sources were imported into the EndNote X9.0 reference manager software (Thompson and Reuters, Philadelphia, PA), and duplicates were removed. Subsequently, two investigators independently (M.R. and H.P.) scrutinized titles, abstracts, and then full-text articles to identify potentially relevant studies for inclusion in the meta-analysis. The following inclusion criteria were applied: (1) observational (cross-sectional, case-control, and cohort) studies that evaluated the association between maternal infection (interested exposure) and development of ID (interested outcome) in children (interested population), (2) studies that utilized universally recognized diagnostic criteria or tests to detect maternal infection or ID in children, involving International Classification of Diseases (ICD) codes or laboratory techniques (serology, molecular or culture methods); (3) studies that provided the odds ratios (ORs), hazard ratios (HRs) or relative risks (RRs) of interested outcomes, or the information was presented to calculate of these risk estimates. Studies were excluded if they failed to quantitatively evaluate of the association between the maternal infection and ID; had not enough data to calculate of an OR and 95% CI; were case reports, case-series, conference papers, systematic reviews, and letters without original data.

### Data extraction and study quality assessment

Two investigators (M.R. and H.P.) independently scrutinized all eligible studies and extracted the following information: first author, publication year, study implementation period, country, diagnostic methodology, type of participants (infection-based or ID-based), study design, the numbers of cases and control subjects, and the prevalence of ID or maternal infection in each of the subject groups. All extracted data were transferred into a standardized Microsoft Excel 2016 spreadsheet (Microsoft Corporation, Redmond, WA). To assess the methodological quality of studies included, we used the Newcastle-Ottawa Scale, which has been recommended by the Cochrane collaboration network [20,21]. This scoring system measures the risk of bias of observational studies based on three dimensions: subject selection criteria (0–4 points), comparability of subjects (0–2 points), and ascertainment of the outcome of interest (0–3 points)—with scores ranging from 0 (low quality) to 9 (high quality). Studies with scores

≥7, 4–6, and ≤3 were considered to be of high, moderate and low quality. All disagreements in study search, study selection and data extraction were resolved in consultation with the lead investigators (M.N.S and A.A.K.).

## Data synthesis and statistical analysis

All statistical analyses were conducted using Stata software (version 17; Stata Corporation, College Station, Texas). The pooled prevalence of the outcome of interest with a 95% confidence interval (95% CI) in each case and control group was estimated using the DerSimonian-Laird random-effects model (REM) [22,23]. Then, the OR and 95% CI were calculated for each individual study with dichotomous data. To assess the association between maternal infection and ID in children, ORs from individual studies were combined to produce a pooled OR and 95% CI, employing the REM with a restricted maximum-likelihood estimator. Between-studies heterogeneity was examined using the Q-test with a $p$-value $< 0.05$ and $I^2$ statistics with a cutoff of $\geq 50\%$ as a significant heterogeneity [24]. In addition, we performed sub-group analyses on type of participants, publication years and diagnostic criteria to identify the sources of heterogeneity and effects of these study characteristics on the outcome of interest. We also performed sensitivity analysis, by iteratively removing each study, to examine the robustness of our pooled estimation for the outcome of interest. Publication bias was evaluated by applying the Begg's and Egger's publication bias methods [25]. A $p$-value of less than 0.05 was considered to be statistically significant.

## Results

### Study characteristics

The PRISMA flow diagram is depicted in Fig 1. Our systematic search yielded 9583 potentially relevant studies. Of those, 2365 articles were excluded following duplicate-removal and 7445 articles were excluded as irrelevant following screening of the title and/or abstract, leaving 43 articles that underwent detailed full-text evaluation according to the inclusion and exclusion criteria. Finally, eight studies examining 60,479 cases and 1,315,183 healthy controls met the criteria for the meta-analysis [15–17,26–30].

The main characteristics of the included studies are summarized in Table 1. All studies had a cohort design, were published between 1998 and 2020, and were performed in three countries: United States (five studies), Sweden (two studies) and China (one study). Four studies were defined as ID-based studies; these studies recruited children affected by ID (12,595 children) as cases and healthy children (736,466 children) as controls and assessed maternal infection in these subjects. On the other hand, four studies were defined as infection-based studies; these studies evaluated the prevalence or incidence of ID in the follow-up of children with- (47,884 children) and without- (578,717 children) exposure to maternal infection. Six studies used International Classification of Diseases (ICD) codes (ICD 8–10) to determine ID and maternal infection, while two other studies used laboratory methods (ELISA and microbial culture) to determine the maternal infection and Wechsler Intelligence Scale for Children (WISC) to identify the children with ID. Based on the methodological quality assessment scores, all studies were deemed to be of high quality.

### Results of meta-analysis

The results of the overall meta-analysis demonstrated that maternal infection significantly increased the risk of ID in children (OR, 1.33; 95%CI, 1.22–1.44) (Table 2 and Fig 2). The between-study heterogeneity was substantial ($\chi^2 = 23.88$; $I^2 = 64.6$). Sensitivity analysis

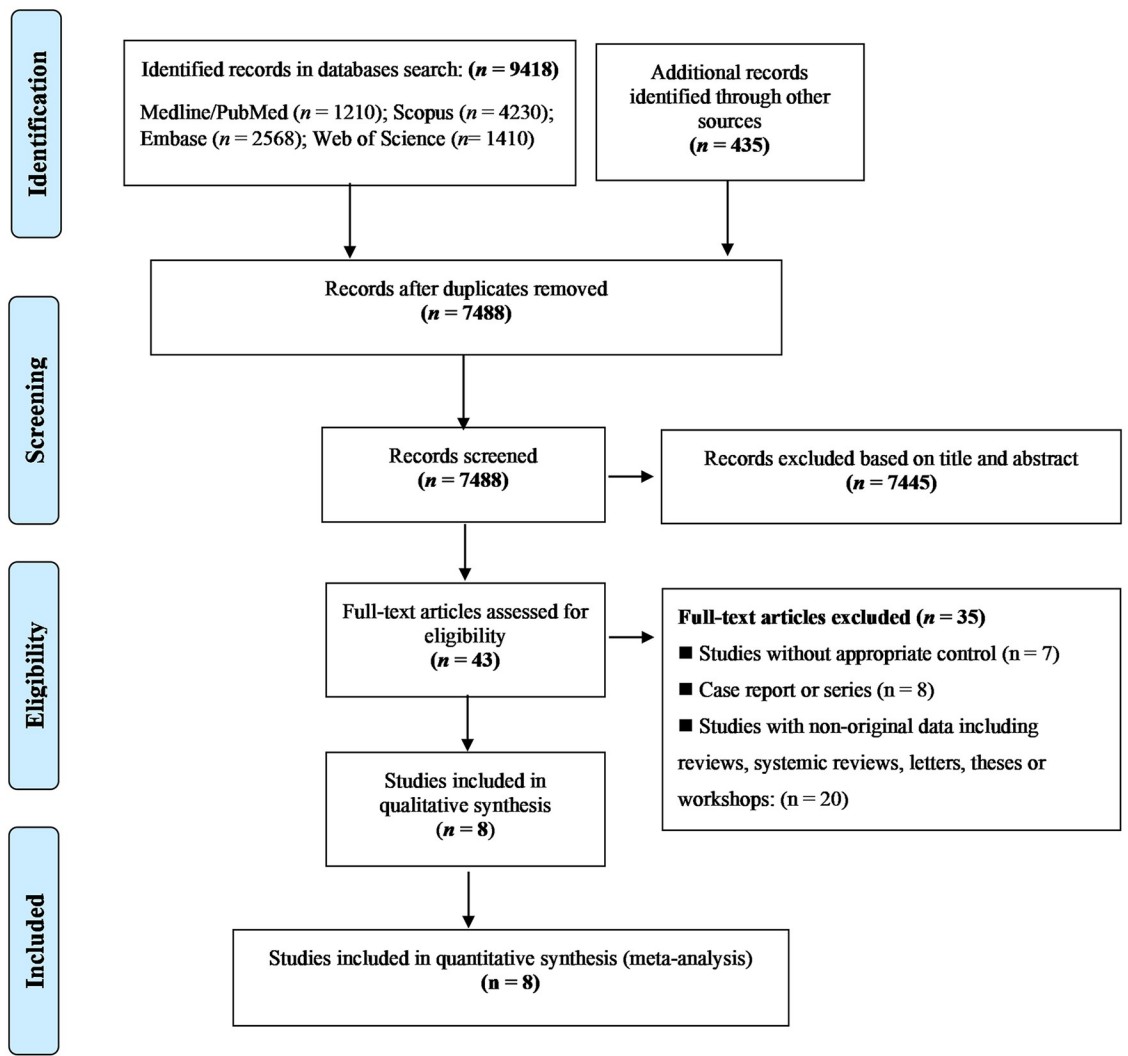

**Fig 1. PRISMA flow chart showing study selection process.**

indicated that exclusion of any individual study did not have a significant influence in pooled OR overall (S2 Fig), revealing high stability of our results. Moreover, a significant association was found in both infection-based- (OR, 1.27; 95%CI, 1.15–1.40) and ID-based- (OR, 1.44; 95%CI, 1.19–1.74) studies (Table 2, Fig 2). A meta-analysis on four studies that had adjusted OR also indicated a significant association (OR, 1.39; 95%CI, 1.28–1.50; $I^2 =$ 0.0) (Fig 3).

Five studies were performed in the United States, and a REM on these studies indicated a significant positive association (OR, 1.34; 95%CI, 1.19–1.50) (S3 Fig). A subgroup analysis of the results according to diagnostic criteria for both ID and maternal infection revealed a significant association when ICD codes (OR, 1.33; 95% CI, 1.19–1.48) were used, while a non-significant association was observed when WISC or laboratory methods (OR, 1.40; 95% CI, 0.92–1.13) were used (S4 and S5 Figs). When the sub-groups were analyzed according to the year of publication, both groups of studies that were published before (OR, 1.23; 95% CI, 1.15–1.31) and after (OR, 1.49; 95% CI, 1.23–1.79) 2010 showed significant positive association (S6 Fig).

**Table 1. Main characteristics of included studies.**

| Studies* | Implementation year | Country | Case | Outcome | Control | Outcome | Method to confirm infection | ID diagnostic criterion |
|---|---|---|---|---|---|---|---|---|
| Camp et al. (1998) [27]# | 1959–1965 | USA | 5244 | 244 | 30476 | 1068 | Microbial culture | WISC |
| Mcdermoot et al. (2000) [28] # | 1995–1998 | USA | 8578 | 673 | 32237 | 2205 | ICD-9 codes | ICD-9 codes |
| Zhang et al. (2007) [29] # | 1997–2000 | China | 49 | 4 | 50 | 1 | ELISA | WISC |
| Mann et al. (2009) [17]§ | 1996–2002 | USA | 5388 | 1366 | 129208 | 27961 | ICD-9 codes | ICD-9 codes |
| Bilder et al. (2013) [16] § | 1994–2001 | USA | 146 | 3 | 16936 | 47 | ICD-9 codes | ICD-9 codes |
| Lee et al. (2015) [26] § | 1984–2011 | Sweden | 2280 | 124 | 471056 | 16352 | ICD-9, and ICD-10 codes | ICD-9, and ICD-10 codes |
| McCarter et al. (2020) [30]§ | 2004–2013 | USA | 4781 | 154 | 119266 | 2774 | ICD-9 codes | ICD-9 codes |
| Brynge et al. (2022) [15]# | 1987–2016 | Sweden | 34013 | 445 | 515954 | 5087 | ICD-8, ICD-9, and ICD-10 codes | ICD-9 codes |

* All studies had retrospective cohort design and were classified as high-quality studies based on Newcastle-Ottawa Scale.

#**Infection-based studies;** these studies evaluated prevalence or incidence of ID in follow-up of children with- and without- exposure to maternal infection.

§**Intellectual disability-based studies;** these studies recruited children affected by ID as cases and healthy children as controls and assessed maternal infection in these subjects retrospectively.

**Abbreviation**; **WISC**, Wechsler Intelligence Scale for Children; **ICD**, International Classification of Diseases.

We assessed the possibility of publication bias using Egger's test and Funnel plot for crude and adjusted ORs. S7 Fig shows a significant publication bias (Egger's test p-value = 0.01 and asymmetrical Funnel plot) for the bivariable association between maternal infection and ID in children (crude OR). However, we did not find any evidence of publication bias in studies that examined the multivariable association (adjusted OR) between maternal infection and ID in

**Table 2. Sub-group analysis of the pooled prevalence and odds ratios for the association between maternal infection and intellectual disability.**

| Variables (number of datasets) | Pooled prevalence of outcome in cases % (95% CI) | Pooled prevalence of outcome in controls % (95% CI) | Odds ratios (95% CI) | Heterogeneity ($I^2$%) |
|---|---|---|---|---|
| **Study design** | | | | |
| Infection-based | 4.68 (1.20–10.08) | 3.03 (0.50–7.31) | 1.27 (1.15–1.40) | 53.6 |
| ID-based | 7.34 (0.45–21.13) | 4.73 (0.36–13.69) | 1.44 (1.19–1.74) | 82.5 |
| **Diagnostic criteria for maternal infection and ID** | | | | |
| ICD-codes | 6.09 (1.28–14.04) | 4.22 (0.98–9.57) | 1.33 (1.19–1.48) | 0.0 |
| Laboratory methods | 4.24 (3.69–4.83) | 3.06 (2.85–3.27) | 1.40 (0.92–1.13) | 99.8 |
| **Publication year** | | | | |
| Before 2010 | 10.59 (3.00–21.88) | 7.46 (0.91–19.09) | 1.23 (1.15–1.31) | 30.4 |
| 2010–2022 | 2.86 (1.13–5.30) | 1.51 (0.48–3.11) | 1.49 (1.23–1.79) | 72.5 |
| **Overall** | 6.07 (1.93–12.20) | 3.86 (1.05–8.25) | 1.33 (1.22–1.44) | 70.7 |

**Abbreviation**; **WISC**, Wechsler Intelligence Scale for Children; **ICD**, International Classification of Diseases.

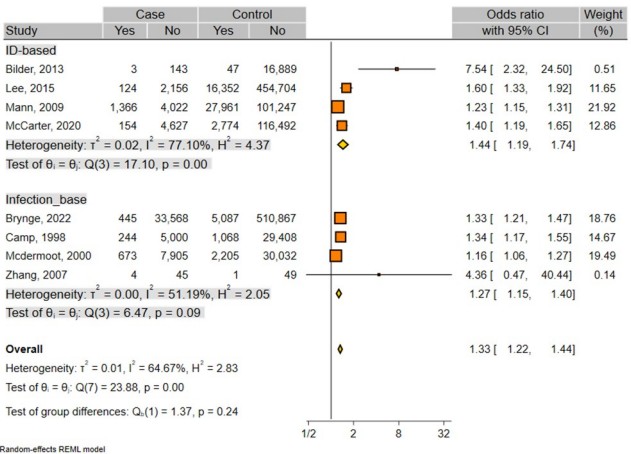

**Fig 2. Forest plot for the association between maternal infection and ID.** Subgroups are based on different type of studies (infection-based- and ID-based- studies). The center of each square represents the OR, the area of the square is the weighted percentage in the meta-analysis and the horizontal line indicates the 95% CI. % weight: weight of each study compared with all the studies.

children (Egger's test p-value = 0.11 and symmetrical Funnel plot; S8 Fig). We also performed a Fill and Trim sensitivity analysis to investigate the effect of publication bias, and found no significant change in the result (pooled OR crude, 1.34; 95% CI, 1.05–1.63) (Fig 4).

## Discussion

ID is a very significant health problem with varied etiology that first develops during pregnancy and is mostly diagnosed in childhood. An improved understanding of the potential risk factors for development of ID may have important public health and clinical implications. The main aim of the present study was to assess whether maternal infection increases the risk of ID in the children. Our meta-analysis findings of eight observational studies concluded that maternal infection during pregnancy could be associated with a significantly increased risk of ID (OR, 1.3; 95% CI, 1.21–1.46) in children. Moreover, almost all subgroup analyses according to type of studies, publication year, and diagnostic criteria also indicated

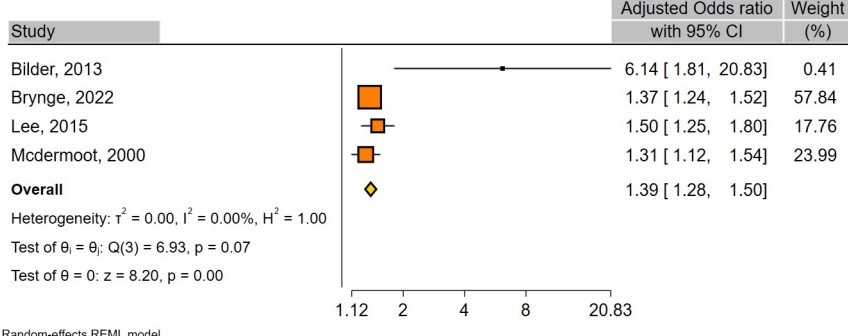

**Fig 3. Forest plot for the association between maternal infection and ID in four studies having adjusted OR.**

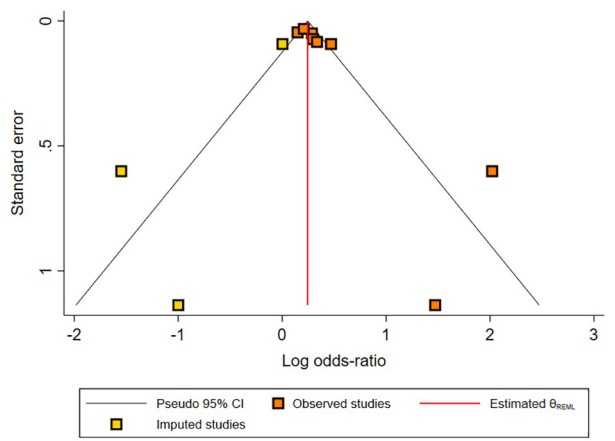

**Fig 4. Fill and Trim sensitivity analysis for publication bias in crude OR.**

a positive association. These findings are in line with previous meta-analyses indicating that maternal infection is associated with increased risk of psychosis, cerebral palsy, and autism spectrum disorders [12–14].

Although precise biological mechanisms have not been clearly elucidated, maternal infection could play an important role in the cause of ID through multiple mechanisms [31]. Experimental/mechanistic studies on the role of maternal infections in development of ID are very rare [15]. But because ID and autism spectrum disorders (ASD) have some common phenotypic and genotypic features [32], and 25–30% of people with ASD are also diagnosed with ID [33], therefore, it has been suggested that infectious organisms can cause ID in the same pathways that they induce ASD [15]. First, it is well-known that some infectious organisms (e.g. *Toxoplasma gondii*, rubella, and cytomegalovirus) can cross the placenta, entering the fetal environment and induce adverse fetal neurodevelopmental outcomes and brain damage [34–37]. Furthermore, proinflammatory mediators/cytokines (TNF, IL-17, IL6, IL-2, and IL-1β) and maternal antibodies induced by host immune system in response to infections, are release into maternal serum and amniotic fluid, could cross the placenta and disrupt fetal neurodevelopment [38–40]. Second, infections during pregnancy, either neurotropic or sexually transmitted infections, are associated with adverse pregnancy outcomes (e.g., low birth weight and preterm delivery), which increase the risk of ID.

The strength of this meta-analysis lies in its rigorous statistical methodology based on cohort studies, large sample size and stratified analyses according to study design, publication year and diagnostic criteria. However, several limitations should be considered regarding this study. Firstly, there was a low number of eligible studies in a few countries (USA, Denmark and China), and there was no study from less developed countries (South America, Africa, South Asia and Middle-East), where the rate of both maternal infection and childhood ID is higher than in developed countries. Second, due to lack of stratified data in original studies, we were unable to perform the subgroup analyses based on specific infections, therefore we were unable to interpret which infection organism had more impact on inducing of ID. Third, there was substantial heterogeneity between studies and a significant publication bias was found by Egger test. According to our analyses publication year and diagnostic criteria might be the source of heterogeneity, although it should be addressed in more well-controlled studies in future.

In conclusion, this meta-analysis study indicated that children exposed to maternal infection during pregnancy are at a greater risk of developing ID. Further multicenter longitudinal studies, especially in less developed countries, that consider the type of infectious agent and more confounding factors are needed to confirm our findings.

## Supporting information

**S1 Checklist. PRISMA 2020 checklist.**
(DOCX)

**S1 Fig. Search strategy.**
(TIF)

**S2 Fig. Sensitivity analysis after each study was excluded.**
(TIF)

**S3 Fig. Pooled OR estimates in subgroups according to the countries for studies assessing the association between maternal infection and intellectual disability (ID) in children.**
(TIF)

**S4 Fig. Pooled OR estimates in subgroups according to the diagnostic criteria for ID in studies assessing the association between maternal infection and ID in children.**
(TIF)

**S5 Fig. Pooled OR estimates in subgroups according to the diagnostic criteria for maternal infection in studies assessing the association between maternal infection and ID in children.**
(TIF)

**S6 Fig. Pooled OR estimates in subgroups according to the publication year for studies assessing the association between maternal infection and ID in children.**
(TIF)

**S7 Fig. Publication bias for studies assessing the association between maternal infection and ID in children, indication a significant publication bias.**
(TIF)

**S8 Fig. Publication bias for four studies having adjusted odds ratio, indication that there is no publication bias in these studies.**
(TIF)

## Acknowledgments

The authors would like to thank Dr. Vahid Fallah Omrani (Calgary University, Canada), for his assistance during the preparation of this manuscript.

## Author Contributions

**Conceptualization:** Mahroo Rezaeinejad, Malihe Nourollahpour Shiadeh, Ali Alizadeh Khatir.

**Data curation:** Mahroo Rezaeinejad, Mehrdad Jafari Tadi, Malihe Nourollahpour Shiadeh.

**Formal analysis:** Seyed Mohammad Riahi, Mehrdad Jafari Tadi, Zahra Geraili.

**Investigation:** Mahroo Rezaeinejad, Seyed Mohammad Riahi, Kimia Behzad Moghadam, Mehrdad Jafari Tadi, Zahra Geraili, Hamid Parsa, Elika Marhoommirzabak, Malihe Nourollahpour Shiadeh.

**Methodology:** Mahroo Rezaeinejad, Seyed Mohammad Riahi, Kimia Behzad Moghadam, Mehrdad Jafari Tadi, Zahra Geraili, Hamid Parsa, Elika Marhoommirzabak, Ali Alizadeh Khatir.

**Software:** Seyed Mohammad Riahi, Zahra Geraili.

**Supervision:** Malihe Nourollahpour Shiadeh, Ali Alizadeh Khatir.

**Validation:** Malihe Nourollahpour Shiadeh, Ali Alizadeh Khatir.

**Visualization:** Malihe Nourollahpour Shiadeh.

**Writing – original draft:** Mahroo Rezaeinejad, Malihe Nourollahpour Shiadeh, Ali Alizadeh Khatir.

**Writing – review & editing:** Malihe Nourollahpour Shiadeh, Ali Alizadeh Khatir.

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
