## [Decision Letter · Decision Letter 0]

19 Jun 2023

PONE-D-23-15712Maternal infection and risk of intellectual disability in children: a systematic review and meta-analysisPLOS ONE

Dear Dr. Nourollahpour Shiadeh,

Thank you for submitting your manuscript to PLOS ONE. After careful consideration, we feel that it has merit but does not fully meet PLOS ONE’s publication criteria as it currently stands. Therefore, we invite you to submit a revised version of the manuscript that addresses the points raised during the review process.

We look forward to receiving your revised manuscript.

Kind regards,

Zemenu Yohannes Kassa, Msc

Academic Editor

PLOS ONE

Journal Requirements:

Additional Editor Comments :

General comments

Your methods and results sections lack clarity and need attention and revision should bed needed.

# Abstract

In lines 30 and 31, the sentence has grammar errors and is rewritten again in an understandable way.

Line 32 and 33 also lack clarity, and it needs revision to make them clear for the reader.

How to assess publications in this meta-analysis, and you should indicate it for a reader in the abstract.

You should clearly show and convey to the readers in the methods part. It lacks clarity and is not written scientifically.

The result part also is not well written. for example, lines 38 and 39.

You can write like …… maternal infection significantly increased ID………

Lines 41 -43 also are not written clearly and need modifications.

# Results

Line 67, please state the laboratory methods they used to identify maternal infections.

In the methods section, you did not explain your outcome variable using PICO or others, which is challenging to catch infection-based and ID findings.

Lines 198 -202, It is confusing the reader, and it needs to rewrite again.

Please revise PRIMA, Figure 1,

Reviewers' comments:

Reviewer's Responses to Questions

**Comments to the Author**

1. Is the manuscript technically sound, and do the data support the conclusions?

Reviewer #1: Yes

Reviewer #2: Yes

2. Has the statistical analysis been performed appropriately and rigorously? 

Reviewer #1: Yes

Reviewer #2: Yes

3. Have the authors made all data underlying the findings in their manuscript fully available?

Reviewer #1: Yes

Reviewer #2: Yes

4. Is the manuscript presented in an intelligible fashion and written in standard English?

Reviewer #1: Yes

Reviewer #2: Yes

5. Review Comments to the Author

Reviewer #1: Rezaeinejad et al., has performed a meta-analysis study on a very interesting topic about Maternal infection and risk of intellectual disability in children. There are some technical and grammatical errors requiring revision from the authors. Please see my comments in details.

1- Please note that Lee et al. (2015) is not an American study, but it is performed on a Swedish population-based register study. Please correct tables and text.

2- The values in the following sentence “Moreover, a significant relationship was found in both infection-based- (OR, 7.10;180 95%CI, 3.50–14.38) and ID-based-(OR, 1.70; 95%CI, 0.99–2.91) studies (Table 2, Figure 2).” are different from those reported in Table 2. Please modify them.

3. Lines 87, 88 “The search strategy was designed and adapted it to each database by a medical library expert (Z.R).” although there is no author with these initials in the authors list.

4. Please use same format to write p value throughout the manuscript.

5. Throughout the manuscript needs to be edited for English language to modify some minor grammatical errors.

Reviewer #2: I read with interest this systematic review and meta-analysis about the maternal infection and risk of intellectual disability in children. The topic is interesting and the paper well written and the methodology correct.

I have some concerns:

1) The term relationship should be replaced by association, as this term is inappropriate for scientific material.

2) The Higgins index (I2), should be added to effect sizes in abstract.

3) There is controversy (typographical error) between Table 2 and results in main text and abstract especially sub-group analysis.

4) It is better to separate the introduction in several paragraphs not two long paragraphs. A paragraph is a group of sentences that convey an idea. Each sentence works together as part of a unit to create an overall thought or impression. A paragraph is the smallest unit or cluster of sentences in which one idea can be developed adequately. Paragraphs can stand alone or function as part of an essay, but each paragraph covers only one main idea. The most important sentence in your paragraph is the topic sentence, which clearly states the subject of the whole paragraph. The topic sentence is usually the first sentence of the paragraph because it gives an overview of the sentences to follow. The supporting sentences after the topic sentence help to develop the main idea. These sentences give specific details related to the topic sentence. A final or concluding sentence often restates or summarizes the main idea of the topic sentence.

5) It is better to write: Medline through PubMed not only PubMed.

6) As, the authors conducted a perfect sub-group analysis, it is better to omit meta-regression.

6. PLOS authors have the option to publish the peer review history of their article (what does this mean?). If published, this will include your full peer review and any attached files.

Reviewer #1: No

Reviewer #2: **Yes: **Mahdi Sepidarkish

---

## [Author Response · Author response to Decision Letter 0]

23 Jul 2023

Dear Dr Yohannes Kassa

Academic Editor

Plos One

Thank you very much for reviewing our manuscript. Comments by you and reviewers were very helpful. We revised the manuscript according to the comments raised by the you and reviewers. All the changes based on the reviewer’s comments are highlighted yellow in the revised manuscript as I explained in follow.

Sincerely yours

Additional Editor Comments :

General comments

Your methods and results sections lack clarity and need attention and revision should bed needed.

# Abstract

In lines 30 and 31, the sentence has grammar errors and is rewritten again in an understandable way.

Line 32 and 33 also lack clarity, and it needs revision to make them clear for the reader.

How to assess publications in this meta-analysis, and you should indicate it for a reader in the abstract.

You should clearly show and convey to the readers in the methods part. It lacks clarity and is not written scientifically.

The result part also is not well written. for example, lines 38 and 39.

You can write like …… maternal infection significantly increased ID………

Lines 41 -43 also are not written clearly and need modifications.

Response: Many thanks for your valuable suggestions. we re-written the abstract section to resolve your concerns.

# Results

Line 67, please state the laboratory methods they used to identify maternal infections.

Response: Many thanks for your valuable suggestions. we added related diagnostic methods.

In the methods section, you did not explain your outcome variable using PICO or others, which is challenging to catch infection-based and ID findings.

Response: Many thanks for your valuable comment. We included those studies that evaluated the association between maternal infection and development of ID in children. So, we expressed the PICO specifically. The interested exposure and outcome are maternal infection and intellectual disorders, respectively. We revised the eligibility criteria as follow: “The following inclusion criteria were applied: (1) observational (cross-sectional, case-control, and cohort) studies that evaluated the association between maternal infection (interested exposure) and development of ID (interested outcome) in children (interested population)…”.

Lines 198 -202, It is confusing the reader, and it needs to rewrite again.

Response: Many thanks for your valuable suggestions. We re-written the paragraph to resolve your concerns.

Please revise PRIMA, Figure 1.

Response: Many thanks for your valuable suggestions. we edited throughout the manuscript to resolve your concerns.

Reviewer #1: Rezaeinejad et al., has performed a meta-analysis study on a very interesting topic about Maternal infection and risk of intellectual disability in children. There are some technical and grammatical errors requiring revision from the authors. Please see my comments in details.

Response: thanks for your valuable comments. We modified all of them. 

1- Please note that Lee et al. (2015) is not an American study, but it is performed on a Swedish population-based register study. Please correct tables and text.

Response: thanks, addressed.

2- The values in the following sentence “Moreover, a significant relationship was found in both infection-based- (OR, 7.10;180 95%CI, 3.50–14.38) and ID-based-(OR, 1.70; 95%CI, 0.99–2.91) studies (Table 2, Figure 2).” are different from those reported in Table 2. Please modify them.

Response: thanks for your detailed attention, we modified them.

3. Lines 87, 88 “The search strategy was designed and adapted it to each database by a medical library expert (Z.R).” although there is no author with these initials in the authors list.

Response: thanks for your detailed attention, we modified them.

4. Please use same format to write p value throughout the manuscript.

Response: thanks, addressed.

5. Throughout the manuscript needs to be edited for English language to modify some minor grammatical errors.

Response: thanks, we edited throughout the manuscript.

Reviewer #2: I read with interest this systematic review and meta-analysis about the maternal infection and risk of intellectual disability in children. The topic is interesting and the paper well written and the methodology correct.

I have some concerns:

1) The term relationship should be replaced by association, as this term is inappropriate for scientific material.

Response: thanks, we replaced relationship by association throughout the manuscript.

2) The Higgins index (I2), should be added to effect sizes in abstract.

Response: thanks, we added I2 for effect size in abstract.

3) There is controversy (typographical error) between Table 2 and results in main text and abstract especially sub-group analysis.

Response: thanks for your detailed attention, we modified them.

4) It is better to separate the introduction in several paragraphs not two long paragraphs. A paragraph is a group of sentences that convey an idea. Each sentence works together as part of a unit to create an overall thought or impression. A paragraph is the smallest unit or cluster of sentences in which one idea can be developed adequately. Paragraphs can stand alone or function as part of an essay, but each paragraph covers only one main idea. The most important sentence in your paragraph is the topic sentence, which clearly states the subject of the whole paragraph. The topic sentence is usually the first sentence of the paragraph because it gives an overview of the sentences to follow. The supporting sentences after the topic sentence help to develop the main idea. These sentences give specific details related to the topic sentence. A final or concluding sentence often restates or summarizes the main idea of the topic sentence.

Response: thanks, we modified the size of paragraphs in introduction. 

5) It is better to write: Medline through PubMed not only PubMed.

Response: thanks, addressed.

6) As, the authors conducted a perfect sub-group analysis, it is better to omit meta-regression.

Response: thanks, addressed.

---

## [Decision Letter · Decision Letter 1]

7 Aug 2023

PONE-D-23-15712R1Maternal infection and risk of intellectual disability in children: a systematic review and meta-analysisPLOS ONE

Dear Dr. Nourollahpour Shiadeh,

Thank you for submitting your manuscript to PLOS ONE. After careful consideration, we feel that it has merit but does not fully meet PLOS ONE’s publication criteria as it currently stands. Therefore, we invite you to submit a revised version of the manuscript that addresses the points raised during the review process.

We look forward to receiving your revised manuscript.

Kind regards,

Zemenu Yohannes Kassa, Msc

Academic Editor

PLOS ONE

Additional Editor Comments:

#Abstract

Background rewrite like this

There is arguing evidence regarding the association between maternal infections during pregnancy and the risk of intellectual disability (ID) in children. This systematic review and meta-analysis are essential to determine and address inconsistent findings between maternal infections during pregnancy and the risk of intellectual disability (ID) in children.

#Methods

Line 33, from when to when and where?

Lines 35-38 you should rewrite again.

#Result

Line 44, ID-based (OR, 1.70; 95%CI, 0.99–2.91; I 2 44 = 51.2) studies. It did not significantly associate with……… why you reported it here?

Subgroup analysis also, it is not clear to readers. What does mean infection-based and ID-based studies?

#Conclusion, it needs gross modification, please pay attention to this section, you should provide a conclusion and recommendation based on your findings.

Keywords: Please use Mesh terms

#Introduction

Line 62,….. responsible for ~10.7 million years lived with disability……..it needs modifications.

Line 66-68, the sentences are out of your objectives. It focuses on maternal behaviour. but not on maternal infectious and remove it.

#Methods: state the diagnostic criteria that you included in your study?

Why did you not use grey literatures?

Did you research using reference of referencing? if yes state it.

Line 113 odds ratios (ORs), hazard ratios (HRs) or relative risks (RRs) of interested outcomes. How to be pooled it.

Line 131, if the score is how many, you included in your review?

Check your PRISMA.

In the table1, you should write types of laboratory used instead of code, for example, PCR, ELISA …… .

ID diagnostics criteria WISC……but not code

Based on your title, you should on “Infection-based studies; these studies evaluated the prevalence or incidence of ID in the follow-up of children with- and without exposure to maternal infection.” But not Intellectual disability-based studies; these studies recruited children affected by ID. because it might occur other factors like alcohol, cigarette, and others. Or modify your titles.

You expected to show OR and RR of each included study in Table 2.

Reviewers' comments:

Reviewer's Responses to Questions

**Comments to the Author**

1. If the authors have adequately addressed your comments raised in a previous round of review and you feel that this manuscript is now acceptable for publication, you may indicate that here to bypass the “Comments to the Author” section, enter your conflict of interest statement in the “Confidential to Editor” section, and submit your "Accept" recommendation.

Reviewer #1: All comments have been addressed

Reviewer #2: All comments have been addressed

2. Is the manuscript technically sound, and do the data support the conclusions?

Reviewer #1: Yes

Reviewer #2: Yes

3. Has the statistical analysis been performed appropriately and rigorously? 

Reviewer #1: Yes

Reviewer #2: Yes

4. Have the authors made all data underlying the findings in their manuscript fully available?

Reviewer #1: Yes

Reviewer #2: Yes

5. Is the manuscript presented in an intelligible fashion and written in standard English?

Reviewer #1: Yes

Reviewer #2: Yes

6. Review Comments to the Author

Reviewer #1: All my concerns were modified; however, I found a mistake in Figure 1. The number of studies after duplicates removal is 5155, but in next step it is 7488. Please correct.

Reviewer #2: The authors have adequately addressed the comments raised in a previous round of review and the manuscript is now acceptable for publication.

7. PLOS authors have the option to publish the peer review history of their article (what does this mean?). If published, this will include your full peer review and any attached files.

Reviewer #1: No

Reviewer #2: No

---

## [Author Response · Author response to Decision Letter 1]

23 Aug 2023

Dear Dr Yohannes Kassa

Academic Editor

Plos One

Thank you very much for reviewing our manuscript. Comments by you and reviewers were very helpful. We revised the manuscript according to the comments raised by the you and reviewers. All the changes based on the reviewer’s comments are highlighted yellow in the revised manuscript as I explained in follow.

Sincerely yours

Additional Editor Comments:

#Abstract

Background rewrite like this

There is arguing evidence regarding the association between maternal infections during pregnancy and the risk of intellectual disability (ID) in children. This systematic review and meta-analysis are essential to determine and address inconsistent findings between maternal infections during pregnancy and the risk of intellectual disability (ID) in children.

R: Many thanks for your valuable corrections. We replaced your corrected sentence.

#Methods

Line 33, from when to when and where?

R: Thanks, we searched databases from their inception up to 15 March 2022. We modified this sentence as follow “The Medline/PubMed, Web of Science, Embase, and Scopus databases were searched from inception up to March 15, 2023, to identify potentially eligible studies.”

Lines 35-38 you should rewrite again.

R: We re-write these sentences again.

#Result

Line 44, ID-based (OR, 1.70; 95%CI, 0.99–2.91; I 2 44 = 51.2) studies. It did not significantly associate with……… why you reported it here?

R: Please accept our apologies, in previous revision we forgotten to modify these values. WE corrected them as follow: Subgroup analysis showed a significant association for both infection-based (OR, 1.27; 95%CI, 1.15–1.40; I2 = 51.2) and ID-based (OR, 1.44; 95%CI, 1.19–1.74; I2 = 77.1) studies.

Subgroup analysis also, it is not clear to readers. What does mean infection-based and ID-based studies?

R: we added following sentence in the method section to clarify infection-based and ID-based studies: “The included studies were divided into two types based on the participants: (1) ID-based studies, which involved children with ID as cases and healthy children as controls and evaluated maternal infection in these participants; (2) infection-based studies, which assessed the prevalence or incidence of ID in the follow-up of children with or without exposure to maternal infection.” Moreover, similar explanation is in main text: lines 169-175.

#Conclusion, it needs gross modification, please pay attention to this section, you should provide a conclusion and recommendation based on your findings.

R: we re-write this section as follow: “Our study suggests that maternal infection during pregnancy could be associated with an increased risk of ID in children. This finding is consistent across different types of studies and diagnostic criteria. However, due to the heterogeneity and limitations of the included studies, we recommend further longitudinal studies to confirm the causal relationship and the underlying mechanisms.”

Keywords: Please use Mesh terms

R: thanks, modified.

#Introduction

Line 62,….. responsible for ~10.7 million years lived with disability……..it needs modifications.

R: thanks, paraphrased it.

Line 66-68, the sentences are out of your objectives. It focuses on maternal behaviour. but not on maternal infectious and remove it.

R: thanks, removed.

#Methods: state the diagnostic criteria that you included in your study?

R: thanks, we modified related sentence as follow: “studies that utilized universally recognized diagnostic criteria or tests to detect maternal infection or ID in children, involving International Classification of Diseases (ICD) codes or laboratory techniques (serology, molecular, or culture methods)”.

Why did you not use grey literatures?

Did you research using reference of referencing? if yes state it.

R: we modified following sentence to response your above both questions: “Furthermore, two investigators (M.J. and M.J.) searched the first 20 pages of the Google Scholar engine using the above keywords and the bibliographies of retrieved articles to find additional relevant studies and gray literature.”

Line 113 odds ratios (ORs), hazard ratios (HRs) or relative risks (RRs) of interested outcomes. How to be pooled it.

R: All studies had a cohort design. All studies reported the raw counts with percentage. So, we calculated the odds ratio from the counts in contingency table. The odds ratio (OR), its standard error and 95% confidence interval are calculated according to Altman, 1991.

We used the following formulae to calculate the odds ratio (OR) and its confidence interval (CI). 

OR = a*d / b*c, where:

a is the number of times both A and B are present,

b is the number of times A is present, but B is absent,

c is the number of times A is absent, but B is present, and

d is the number of times both A and B are negative.

To calculate the confidence interval, we use the log odds ratio, log(or) = log(a*d/b*c), and calculate its standard error:

se(log(or)) = √1/a + 1/b + 1/c +1/d

The confidence interval, ci, is calculated as:

ci = exp(log(or) ± Zα/2¬*√1/a + 1/b + 1/c + 1/d),

Where Zα/2 is the critical value of the Normal distribution at α/2 (e.g. for a confidence level of 95%, α is 0.05 and the critical value is 1.96).

Additionally, four studies reported adjusted effect size (three studies reported odds ratio and one study reported hazard ratio). As you know there is no precise formula to convert HR to RR or OR. In short, under very specific conditions, the HR and RR will coincide when everyone has complete follow-up (no dropouts or censoring). It may be possible to compute an approximate RR by inspection of the Kaplan-Meier curves by finding the group-specific failure (or event) rates at a fixed time-point, since these are probabilities. In practical meta-analysis the researchers suppose the RR and HR identical. Odds ratios approximate risk ratios when the outcome under consideration is rare but can diverge substantially from risk ratios when the outcome is common. Logistic regression is used frequently in cohort studies and clinical trials. When the incidence of an outcome of interest is common in the study population (>10%), the adjusted odds ratio derived from the logistic regression can no longer approximate the risk ratio. The more frequent the outcome, the more the odds ratio overestimates the risk ratio when it is more than 1 or underestimates it when it is less than 1. In this meta-analysis, we considered the hazard ratio and the odds ratio to be the same.

Line 131, if the score is how many, you included in your review?

R: as mentioned in the result section: “Based on the methodological quality assessment scores, all studies were deemed to be of high quality.”

Check your PRISMA.

R: according to comment by one of the reviewers we modified PRISMA flowchart. 

In the table1, you should write types of laboratory used instead of code, for example, PCR, ELISA …… .

ID diagnostics criteria WISC……but not code

Response: Please note that International Classification of Diseases (ICD) are most validated criteria for diagnosis of many diseases. We included related information if there were in the included studies.

Based on your title, you should on “Infection-based studies; these studies evaluated the prevalence or incidence of ID in the follow-up of children with- and without exposure to maternal infection.” But not Intellectual disability-based studies; these studies recruited children affected by ID. because it might occur other factors like alcohol, cigarette, and others. Or modify your titles.

Response: please note that our main was evaluate the association between maternal infection and intellectual disability. According to our consultant with epidemiologists as well as our previous experiences we can include both infection-based and ID-based of studies. However, we are agreeing with you that our title may not be suitable, therefore to resolve your concern we modified the title as follow “The association between maternal infection and intellectual disability in children: a systematic review and meta‐analysis” 

You expected to show OR and RR of each included study in Table 2.

 Response: Table 2 is related to sub-group analyses. Related OR for each study is presented in Figure 2.

---

## [Editor Report · Decision Letter 2]

11 Sep 2023

PONE-D-23-15712R2The association between maternal infection and intellectual disability in children: a systematic review and meta‐analysisPLOS ONE

Dear Dr. Nourollahpour Shiadeh,

Thank you for submitting your manuscript to PLOS ONE. After careful consideration, we feel that it has merit but does not fully meet PLOS ONE’s publication criteria as it currently stands. Therefore, we invite you to submit a revised version of the manuscript that addresses the points raised during the review process.

We look forward to receiving your revised manuscript.

Kind regards,

Zemenu Yohannes Kassa, Msc

Academic Editor

PLOS ONE

Journal Requirements:

**Additional Editor Comments:**

General comments

You should be expected to follow PLOS ONE guidelines.

Abstract

See punctuation.

Lines 27,32, 45, 53

line 46: what is REM?

Discussion

Line 229 -234, see again; this paragraph explains ID related to behaviours.

---

## [Author Response · Author response to Decision Letter 2]

12 Sep 2023

Dear Dr Yohannes Kassa

Academic Editor

PLOS One

Thank you very much for reviewing our manuscript. Comments by you and reviewers were very helpful. We revised the manuscript according to the comments raised by the you and reviewers. All the changes based on the reviewer’s comments are highlighted yellow in the revised manuscript as I explained in follow.

Sincerely yours

Journal Requirements:

R: Thank you very much for your inquiry. We have diligently reviewed all the articles referenced in this paper, utilizing both PubMed and the Retraction Watch website (http://retractiondatabase.org/). Regrettably, we did not uncover any retracted papers among the citations. Should you come across any retracted articles, we would greatly appreciate it if you could provide the title of the article so that we can appropriately update our reference list.

Additional Editor Comments:

General comments

You should be expected to follow PLOS ONE guidelines.

Abstract

See punctuation.

Lines 27,32, 45, 53

R: Many thanks for your detailed attention, we followed PLOS ONE guidelines in abstract.

line 46: what is REM?

R: Thanks, REM is an abbreviation for Random-effects models. We added it in Abstract

Discussion

Line 229 -234, see again; this paragraph explains ID related to behaviours.

R: Many thanks for your valuable comments, all your concerns were modified. We deleted this sentence and modified related sentences.

---

## [Editor Report · Decision Letter 3]

18 Sep 2023

The association between maternal infection and intellectual disability in children: a systematic review and meta‐analysis

PONE-D-23-15712R3

Dear Dr. Malihe Nourollahpour Shiadeh,

We’re pleased to inform you that your manuscript has been judged scientifically suitable for publication and will be formally accepted for publication once it meets all outstanding technical requirements.

Kind regards,

Zemenu Yohannes Kassa, Msc

Academic Editor

PLOS ONE
---

## [Editor Report · Acceptance letter]

25 Sep 2023

PONE-D-23-15712R3 

The association between maternal infection and intellectual disability in children: a systematic review and meta‐analysis 

Dear Dr. Nourollahpour Shiadeh:

I'm pleased to inform you that your manuscript has been deemed suitable for publication in PLOS ONE. Congratulations! Your manuscript is now with our production department. 

Kind regards, 

on behalf of

Dr. Zemenu Yohannes Kassa 

Academic Editor

PLOS ONE